**Data Availability Statement:** The data are all contained within the manuscript. Further communication with the first author can be made if need more data. There are no supporting

# Health care managers' perspectives on workforce licensing practice in Ethiopia: A qualitative study

Eshetu Cherinet Teka[1]*, Meron Yakob Gebreyes[1]*, Endalkachew Tsedal Alemneh[1]*, Biruk Hailu Tesfaye[1], Firew Ayalew Desta[2], Yohannes Molla Asemu[2], Ermias Gebreyohannes Wolde[1], Wondimu Daniel Ashena[1], Samuel Mengistu[3], Tewodros Abebaw Melese[1], Fikadie Dagnew Biset[1], Bezawit Worku Degefu[1], Bethlehem Bizuayehu Kebede[1], Tangut Dagnew Azeze[1], Wudasie Teshome Shewatatek[1], Melese Achamo Seboka[1], Abera Bezabih Gebreegzi[1], Mekonnen Desie Degebasa[1], Tsedale Tafesse Lemu[1], Yeshiwork Eshetu Abebe[1], Matias Azanaw Alayu[1], Fatuma Ahmed Ebrahim[1], Eden Workneh Sahlemariam[1], Genet Kifle woldesemayat[1], Hailemaryam Balcha Admassu[1], Bethlehem Shikabaw Chekol[1]

1 Health Professionals' Competency Assessment and Licensing Directorate, Ministry of Health, Addis Ababa, Ethiopia, 2 Health Work Force Improvement Program, Jhpiego, Addis Ababa, Ethiopia, 3 Health Work Force Improvement Program, Ethiopian Medical Association, Addis Ababa, Ethiopia

* eshecher2005@gmail.com (ECT); meron@moh.gov.et (MYG); endalkachew@moh.gov.et (ETA)

## Abstract

### Background

Professional licensing bodies are valuable sources for tracking the health workforce, as many skilled health-care providers require formal training, registration, and licensure. Regulatory activities in Ethiopia were not effectively implemented due to poor follow-up and gaps in skilled human resources, budget, and information technology infrastructure.

### Objective

The aim of this study was to explore and describe the lived experiences and challenges faced by health care managers in health professionals' licensure practices in Ethiopia.

### Methods

A cross-sectional study design with a phenomenological approach was employed between March 26 and April 30, 2021, to collect qualitative data. We conducted in-depth interviews with a total of 32 purposively selected health system managers. An interview guide was prepared in English, translated into Amharic, and then pretested. Audio recorded data was transcribed verbatim, translated, and analysed manually by themes and sub-themes. A member check was done to check the credibility of the result.

### Results

The data revealed four major themes: awareness of licensing practices, enforcement of licensing practices, systems for assuring the quality of licensing practices, and challenges to

information files uploaded alongside our paper at this time. In this qualitative study, reflections of participants were analysed from verbatim transcriptions. We did not include any quantitative data. Therefore, quantitative expressions such as means, standard deviations and other measures were not reported. Similarly, graphs were not built; points were not extracted from images for analysis. Since our study is original and had not utilized data set from others, there are no ethical, legal, or third-party restrictions.

**Funding:** There was no funding for this study. The study was conducted by the voluntary contribution of authors at any of the research and manuscript processing. No one of the authors received salary since there was no funder. The authors received no specific funding for this work.

**Competing interests:** There is no competing interest among the authors.

**Abbreviations:** CEUs, Continuous Educational Units; COC, Center of Competence; CPD, Continuous Professional Development; EPHI, Ethiopian Public Health Institute; FMHACA, Food, Medicine, and Healthcare Administration and Control Authority; HMIS, Health Management Information System; IRB-, Institutional Review Board; JEG, Job Evaluation and Grading; MD, Medical Doctor; NMLE, National Medical Licensure Examination; PhD, Philosophic Doctorate; RHIS, Routine Health Information System; SD, Standard Deviation.

licensing practices. Lack of awareness among managers about health workforce licensing was reported, especially at lower-level employers. Regulators were clear on the requirements to issue a licence to the health workforce if they are competent in the licensing exam, while human resource managers do not emphasise whether the employees have a licence or not during employment. As a result of this, non-licenced health workers were employed. Health care managers mentioned that they did not know any monitoring tools to solve the issue of working without a licence. Fraudulent academic credentials, shortage of resources (human resources, finance, equipment, and supplies), and weak follow-up and coordination systems were identified as main practice challenges.

## Conclusions

This study reported a suboptimal health professionals' licensing practice in Ethiopia, which is against the laws and proclamations of the country that state to employ all health workers only with professional licenses. Challenges for health professionals' licensing practice were identified as fraudulent academic credentials, a shortage of resources (HR, finance, equipment, and supplies), and a weak follow-up and coordination system. Further awareness of licensing practices should be created, especially for lower-level employers. Regulators shall establish a reliable digital system to consistently assure the quality of licensing practices. Health care managers must implement mechanisms to regularly monitor the licensing status of their employees and ensure that government requirements are met. Collaboration and regular communication between regulators and employers can improve quality practices.

## Background

Globally, particularly in the private sector, where there is little or no administrative regulation, the public is served by incompetent health care workers [1]. The health workforce regulators in low- and middle-income countries (LMICs) lack the capacity to effectively implement and maintain the system [2,3]. Regulatory agencies in Ethiopia were not effectively discharging their responsibilities due to poor follow-up and inadequate professional registration and licensure practices [4]. Registration and licensing practices in regions are not supported by an automated human resource management information system. Besides, the subnational regulators lacked the capacity to implement registration and licensing functions properly with gaps in skilled human resources, budget, and information technology infrastructure [5].

The Ministry of Health, Ethiopia focuses on regulating health professionals as a strategic initiative in its Health Sector Transformation Plan-II (HSTP-II) [6]. Health professionals' Competency Assessment and Licensure Directorate (HPCALD) has been developing and administering National Licensing Examination (NLE) as a health workforce regulation mechanism since 2019. This is to assess the competence of new bachelor degree graduates of health workforce and to provide licences only for competent health professionals [7–9]. Similarly, the Ministry of Health and the Ministry of Education have developed a training standard and assessment tool for graduates from technical and vocational level since 2009. Like the Bachelor of Science (BSc) graduates, the technical and vocational level graduates have been expected to pass an occupational competence assessment as a prerequisite of issuing a license and workforce entry. Competent Bachelor of Science (BSc), technical and vocational level graduates will receive the license from the regional health bureaus [10].

Ethiopia adopted various health profession regulatory frameworks, though their implementation has fallen behind, partly due to the inadequacy of the number, experience, and qualifications of human resource managers [11]. Ethiopia, as a Sub-Saharan African (SSA) country, has a shortage of human resources for health (HRH). In some countries, this shortage is so severe as to constitute a crisis in the health sector and to have a direct effect on the achievement of the sustainable development goals (SDGs) and particularly the realisation of universal health coverage (UHC) [12].

Beyond anecdotal information, little is known about health professionals' licensing practices and its challenges in LMICs, including Ethiopia. Therefore, this study aimed to explore the perspectives and challenges of health work force licensing among health care managers in Ethiopia.

## Methods

### Study design and setting

A cross-sectional study design with a descriptive phenomenology approach was employed to collect qualitative data in a naturalistic setting. This study was conducted in Ethiopia, a country in Africa with an estimated population of 110 million [13]. After re-structuring in 2020, there are 10 administrative regions as well as the two city administrations of Addis Ababa and Dire Dawa. Regions are subdivided into 98 zones and further into 923 woredas, which in turn are divided into the lowest administrative unit, the kebele, which has around 5000 inhabitants [14]. At each administrative level, the health workforce is managed by a human resources manager and a regulatory manager. Each upper administrative level is responsible for the management, coordination, and distribution of technical support to the lower level [15]. All health care cadres are expected to handle licences during recruitment. Currently, the major health cadres of the country that are assessed by licensure exam to have a license are medicine, laboratory, nursing, public health officer, midwifery, pharmacy, anaesthesia, radiology, dental medicine, psychiatry and nursing specialities [16].

### Study population and sampling procedure

The target population for this study were health care managers working at different administration levels in Ethiopia. A total of 32 purposively selected human resources and regulatory body managers were interviewed between March 26, 2021, and April 30, 2021.

The reason behind using purposive sampling in this study is to select a sample of health care managers who can most likely provide the perspective regarding the health work force licensing practice in Ethiopia needed to answer the research question. Purposive sampling is often used when the population of interest is small, and we used purposive sampling because we wanted to focus in depth on relatively small samples. We used it because we wanted to access these key informant health workforces that shared their experiences with us. Participants were recruited along the flow of the Ethiopian health administration system, which included 24 participants from regional level (12 regional-level human resource managers and12 regional-level regulatory managers), 4 participants from Zonal level(4 zonal-level human resource managers, and 4 participants from district level (4 district-level human resource managers). The twelve regions including city administrations included for the study were Tigray, Afar, Amhara, Oromia, Somali, Benishangul Gumuz,Gambela, Sidama, South Nations Nationalities and peoples of Ethiopia(SNNP),Dire Dawa, Addis Abaaba, and Harari. In this study, twelve regions a, four zones and four districts were selected. The twelve regions were selected because there were 12 regional states in Ethiopia during the data collection period. The four zones and four districts were selected purposively based on their burden of

licensing practice problems based on the national data. To prevent unauthorised access or use, data have been stored securely. Interviewees who had less than one year of work experience in their position were excluded from the study.

## Data collection and quality assurance

A qualitative interview guide was prepared by the authors and a team who were involved in the study. The guide was comprehensive and covered all parts of the objectives we wanted to explore. It encouraged participants to share their experiences in their own words and in their own way without being forced by categories or classifications imposed by the interviewer. It was prepared in English and translated into Amharic. The tool was back-translated into English by an expert to ensure its consistency. The interview guide was pretested and refined to improve its clarity. Sixteen of the authors (eight females and eight males), who were public health professionals in their professions, competency assessment and licensure experts in their occupations, and experienced in qualitative research data collection methods, were recruited for the data collection activity. There was no special relationship between data collectors and participants. Data collectors were trained on techniques of qualitative data collection (interviewing, note-taking, and probing) before deployment to data collection.

Ahead of the interview, a schedule was arranged with participants for the data collection. Data were collected using a semi-structured face-to-face individual interview. Probing questions were applied to find out a detailed idea. The interviews were conducted at the participant's workplace using Amharic since all the study participants were known to have the skill of the language. No one else was present besides the participants and researchers during the interview to enhance the freedom of the participant to give ideas freely. For an interview, two data collectors were assigned: one interviewer and the other note-taker.

The investigators supervised the data collection process thoroughly. Data collection was started at the regional level, then at the zonal level, and finally at the district level. Human resource managers and regulatory managers at the regional level, human resource managers at the zonal level, and human resource managers at the district level were interviewed. Interviews lasted from 45 to 55 minutes. Data saturation was discussed and ended at the 32nd interview. There was no refusal or dropout from participating in the study. Audio recordings were transcribed verbatim into Amharic by the investigators and translated into English on the same day to avoid the loss of details. Four (10%) of the transcripts were cross-checked with the audio for completeness and accuracy. Besides these, a member check was done by sharing the summary result among four respondents to check the credibility of the result [17].

## Data management and analysis

We used thematic content analysis to analyze the result. Verbatim transcripts were read and investigators agreed upon a coding frame. Coding was done by eight of the investigators (authors). Codes were given for emerging ideas. We ensured variation in coding and analyzing the data by having multiple coders to identify discrepancies and maintain consistency. Inter-coder reliability is established by comparing results from multiple coders. Disagreements are identified and resolved to maintain consistency. A senior researcher (among the authors) reviewed the coding and analysis process to ensure it aligns with the research question and results are reliable and valid. Then, codes were categorized into broader thematic areas. Quotes were extracted and collected under identified codes. There were four themes namely awareness on licensing practices, enforcement of licensing practices, Systems for assuring the quality of licensing practices, and challenges to licensing practices. Under the theme named challenges of licensing practices, there were three sub themes named as fraudulent academic credentials,

shortage of resources (HR, finance, equipment, and supplies) and weak follow up and coordination system. A report on qualitative results was prepared. Outliers and contradictory findings were checked and consensus was reached on distribution of key themes [18]. In this study, health care managers were represented by regulatory managers and human resource managers in Ethiopia.

## Ethics approval and consent to participate

Ethical approval for this study was obtained from the Ethiopian Public Health Institute, Institutional Review Board, with certificate ref. no. EPH 6.13/34.E. Before starting data collection, permission and support letters were collected from the Ministry of Health, regional health bureaus, and zonal health departments. Written informed consent was obtained from each study participant after a clear explanation of the study objectives. Privacy and confidentiality of participants' were kept anonymous, including their right to withdraw at any time. The name of the study participant in relation to the finding was not disclosed at any time. All methods applied in this study were performed in accordance with the relevant guidelines and regulations. The research will be published in an international, reputable open-access journal.

## Results

### Socio demographic characteristics

The mean age of key informants was 38.3, with a range of 20 to 60 years. Twenty-four (75%) of the participants were from a regional level. Twelve (37.5%) participants worked for 6–10 years in their current position, while the least two (6.25%) worked for 16–20 years. Twenty-eight (87.5%) participants never received relevant formal training (Table 1).

### Themes and subthemes

There were four themes namely awareness on licensing practices, enforcement of licensing practices, Systems for assuring the quality of licensing practices, and challenges to licensing practices. Under the theme named challenges of licensing practices, there were three sub

**Table 1. Socio demographic characteristics of study participants, Ethiopia, 2021.**

| Characteristics | Number | percentage |
|---|---|---|
| **Age in years** | | |
| 20–29 | 2 | 6.3 |
| 30–39 | 18 | 56.2 |
| 40–49 | 9 | 28.1 |
| 50–59 | 2 | 6.3 |
| 60–69 | 1 | 3.1 |
| **Working level of participants** | | |
| Regional Health Bureau | 24 | 75 |
| Zonal Health Department | 4 | 12.5 |
| District Health Office | 4 | 12.5 |
| **Work experience with current position** | | |
| 1–5 | 11 | 34.3 |
| 6–10 | 12 | 37.5 |
| 11–15 | 7 | 21.8 |
| 16–20 | 2 | 6.25 |
| **Ever received formal RHIS/HMIS/M&E training** | | |
| Yes | 4 | 12.5 |
| No | 28 | 87.5 |

themes named as fraudulent academic credentials, shortage of resources (HR, finance, equipment, and supplies) and weak follow up and coordination system.

## Awareness of licensing practices

More than half of the health care managers replied that they had no clear information about health professionals' licensure exams. They simply heard about it without a detailed understanding.

"I do not have much clarity about it. I heard information disseminated from the zonal level, and even I did not know when it was started." (District human resource manager)

Some health care managers even have not heard about health professionals' licensure exam, that was one of the criteria to issue a licence according to Ethiopian law.

"I have never heard about health professionals' licensure exams or even who is responsible for organising it.But I have a little understanding about the exam organised by regions." (Zonal human resource manager)

## Enforcement of licensing practices

Respondents mentioned that they tried to pay serious attention to important fulfilments while issuing a license. They made an activity of authentication of documents and asked the applicant to fulfil the mandatory criteria.

"----firstly, the applicant to be issued a licence has to apply and bring credentials with the original and copy starting from grade eight up to the highest level of education, and we cross-check the copy with the original. We will check the medical; he or she will pay for the service, bring two passport-size photos, an identity card, and all the other requirements. Finally, he or she will be given the licence." (Regional FMHACA head)

As per the response of health care managers, employed health professionals were asked to bring additional documents to issue a licence other than the documents asked for the newly graduated health professionals.

"serving for two years and bringing a support letter from their working institution are also criteria for having a licence if they have working experience." (District human resource manager)

Health care managers pointed out that the regulatory body is given the mandate to issue a license. Relicensing currently requires continuous professional development (CPD).

"Licence is already given by the regulatory body. Health professionals will be relicensed if they score 30 continuous educational units (CEUs) in a year, and the initiative has begun this year. "(Regional Human Resource Manager)

Respondents replied that a professional licence was issued for a limited number of professions at the BSc level after taking the licensure examination and being competent.

"The professions that have taken the exam were seven. They can be listed as medicine, anaesthesia, nurses, midwives, etc., because they are directly related to the main job, but all health professionals need a icense." (Regional human resource manager)

Though the issue of licensing is too critical as per the ministry's direction, health care managers have been reluctant to take action against those health professionals who breach the law and regulations.

"For those professionals working without licence, we didn't start to take any measures. We simply communicate with zonal-level authorities." (Regional FMHACA manager)

## Systems for assuring the quality of licensing practices

Some Health care managers mentioned that they did not know any monitoring tool to solve the issue of working without a license.

". . .We know that there are health professionals who are working without license. I do not know how to make all working health professionals have a license." (Regional human resource manager)

Most Health care managers didn't give much concern about the licensing status of health professionals. They did not have a plan to inspect the license at the health care sectors/ facilities.

". . . We don't monitor the availability of license for each profession. We don't have schedule for it for any intervention." (Regional human resource manager)

There were few Health care managers that could monitor routinely the availability of a license. They informed health professionals without a license to avail in their individual record.

"To make sure the availability of license for a health professional on the job, we audit their recruitment document/ file. Those professionals without license were informed to bring it soon." (Zonal human resource manager)

Few of the Health care managers replied that they monitor the availability of license by digital system (database) that makes it easy and fast.

"Often, there may be unlicensed health professionals. We enter their file in a computer. By using it, we renew the license as per the exact date'.'(Regional regulatory manager)

"We have a human resource manager is-data system to work with their license. When the license is out of date, the system announces by itself." (Regional human resource manager)

About four of the health care managers reported that health professionals could get promotion in their professional career every 3 years that needs basic criteria to have an active license. ". . .especially this year, professional career is one of the criteria to have a professional license. Anyone without a licence can't get career promotion. To process Job Evaluation and Grading (JEG), license is mandatory." (Regional human resource manager)

Health care managers expressed that health professionals were revoked their license because of different malpractices. They heard it from the upper level by circular letter.

". . . . . . .firstly, a person with a revoked license has been announced by BSC meeting platform. Secondly, it is announced by circular letter from the upper administrative level." (Regional human resource manager)

### Challenges of licensure practice

Respondents mentioned several reasons for poor licensing practice, presented below: fraudulent academic credentials, shortage of resources (HR, finance, equipment, and supplies), and weak follow-up and coordination systems.

## Fraudulent academic credentials

Health professionals had forged professional licence that seemed prepared by the regulatory body. Those who were in charge were under the control of the court.

> "Many forges are increasing in number. There are many cases under the process of the court. The Food and Drug Authority (FDA) is controlling the process" (regional human resource manager)

> ". . . now, there are many private health institutions. But it is difficult to say that all health professionals have a license. A forged licence was made by the titter of our former authorised colleague who passed away" (Regional FMHACA manager).

According to the responses of health care managers, employment by forged licence was done mainly at the district level. Regional-level regulatory bodies could distinguish the forged licence from the original.

> "At the district centres, there is a situation of employment by forged license. We can easily identify whether the licence is forged or not when they are bringing it to us." (Regional regulatory manager)

In two regions, it was reported that health professionals commonly got the wrong professional nomenclature, making them the most senior without deserving it.

> "—the gap here is that the newly graduated health professional was named 'expert professional' [a nomenclature reserved for those who practiced for nine years] without deserving it due to the lack of accountability of the regulatory personnel who issued the license." (Regional human resource manager)

## Shortage of resources (human resources, finance, equipment, and supplies)

The imbalance between available human resources and the workload, especially in larger regions, was repeatedly mentioned as a challenge. Besides, they were limited in their ability to practice health professionals' licensing effectively due to a lack of transportation.

> "There is a shortage of human resources, especially when we compare the area of the region. Moreover, there is a shortage of vehicles for transport. Both the two are unsolved problems still now." (Regional regulatory manager)

Health care managers indicated that they could not create awareness among health professionals because of budget deficiencies and a lack of vehicle for transportation.

> ". . .for employed health professionals, we do not have a budget and transportation to create awareness about licensing." (Regional regulatory manager)

Respondents in some regions replied that health professionals could not afford to cover expenses such as transportation cost when they were going to a regional regulatory body from their local area to take a license, and even they could not access transportation.

"Some professionals coming to take a licence from district level suffered from transportation inaccessibility and shortage of transportation fee." (Regional regulatory manager)

Health care managers reported that they lacked information and communication materials and trained professionals.

"We have only one computer. It needs frequent maintenance, and we cannot perform our routine activities. The printer is non-functional, and we are going to another office to print. We do not have internet service, a scanner, or an information technology officer." (District human resource manager)

## Weak follow-up and coordination system

There was poor follow-up to search for those employed health professionals who did not take the Centre of Competency (COC) exam in 2004/2005. Besides, there was a gap in the identification of individuals who did not take the exam.

"There were health professionals who started jobs around 2013. So, during that time, there was a gap to follow as to whether they took the COC exam or not. Moreover, there was also a gap in checking the data of COC-taker health professionals from non-takers."(Regional human resource manager)

Health care managers replied that they had an interest in seeing if the licensing of health professionals could be decentralised to zonal and district levels. But this could not get an answer from the superior authorities.

". . .as it is known, the authority to issue a licence is practiced only at the regional level, and it is not decentralised to the zonal and district levels. Despite our request to decentralise the structure to the zonal and district levels, it has not gotten a solution until now." (Regional regulatory manager)

According to health care managers, the Ministry of Health gave direction to all regions to practice health professionals' licensing uniformly, despite the actual implementation differences from one region to the other.

The other issue is the difference from one region to another when issuing a license. When we went in the right direction, they asked us, "What is the difference between your rule and that of another region?" (Regional regulatory manager)

A district human resource manager reported that the human resource department and regulatory body lacked coordination when licensing health professionals, despite working together on materials and supply control.

". . .. Most of the time, we worked co-ordinately on controlling materials but not on health professionals' licensing." (District human resource manager)

## Discussion

This phenomenological qualitative investigation explored and described the lived experiences of 32 health care managers on health professionals' licensing practice and its main challenges in Ethiopia. Not including the perspectives of non-managers or practitioners in this study might lead to a limited understanding of the licensing practice at hand and lack of diversity in the data collected and analysed. The data analysis identified four themes: awareness of licensing practices, enforcement of licensing practices, systems for assuring the quality of licensing practices, and challenges to licensing practices. The challenges of licensing practices were categorized in to three sub-themes: fraudulent academic credentials, shortage of resources (HR, finance, equipment, and supplies), and weak follow-up and coordination system. The findings of this study will hopefully influence policymakers, program managers, development partners, health care managers, and health workers in the future to improve efforts towards improving the practice of health professionals' licensing.

According to this study, health care managers hired health professionals without a license which was against the laws and proclamations of the country [19] A sim.ilar finding was observed in Cambodia [20] and India [21]. This was due to the fact that health professionals' licensing was not a priority; poor awareness about health professionals' licensing, corruption, shortage of the regulatory work force and negligence to obey the regulation played the role.

A gap in regular monitoring of the licensing status of health professionals was observed. A similar finding was observed in the Lao People's Democratic Republic [22]. This was because of the shortage of the health work force at regulatory offices; thus, they are busy with office work and consider the licensing status of employed health professionals a secondary issue.

Unlike the regulatory managers, human resource managers at all levels, predominately district and zonal levels, faced a knowledge gap on the mechanism of tracking forged license. A similar condition was expressed in the human resources for health strategy 2016–2025 [7]. This was due to a lack of training for human resource managers because of the insufficient budget allocated for capacity building, workforce motivation, and performance support. This gap could be tackled by providing on-the-job training about health professionals' license because it can boost their knowledge, skills and could motivate them.

Shortage of resources such as trained human resources, budget, and information communication materials were the most important determinants that could affect the licensing practice at all levels of the health system. This finding is supported by World Health Organization that stated the problem is expanding to developed countries. This is a big challenge in this era in which the number of private health facilities is expanding, the emergence of new employment opportunities, humanitarian crises, and increasing patient demand and expectation for high-quality service. This needs a short-term and long-term plan that involves development partners and those who have an interest in the area.

In most of the regions, there was a loose coordination between the human resource department and regulatory body. This finding is supported by a study conducted in India [21]. This poor coordination played a role in hindering effective implementation of licensing practice. This was because human resource managers provide license to employees without fulfilling essential regulatory requirements, and the human resource managers did not communicate with the regulatory body about their supervision results that needed the consultation of the regulatory body. This could be solved by sensitising them to have a culture of working together since the issue of health professional licensing is their mutual task.

Moreover, there was an observed regional variation in the implementation of health professionals' licensing practices. This finding is supported by a study conducted to compare regional regulation variation among the USA, England, Canada, and Australia [23]. It was

because the regional regulatory bodies customised the rules and regulations of the ministry as per their regional context. This implementation variation is challenging, and the source of complaints among candidates could have a negative impact on their psychology. This can be solved by having a discussion between the Ministry of Health and regional regulatory directors.

In summary, a strong monitoring system is essential to be developed to close the gaps between unlicensed working health professionals, forged license, and undeserved naming.

## Strengths and limitations

This study has strengths as it included samples from all regions of Ethiopia (except areas with security problems during the data collection period). Besides, the study included participants from the regional level to the district level and aimed to explore practices and challenges at all levels of the health system. The in-depth interviews were conducted by experienced public health professionals. However, the study had limitations as it did not include key informants at the national level or in the private sector.

## Conclusions

This study reported a suboptimal health professionals' licensing practice in Ethiopia. Health care managers did not have regular monitoring of employed health professionals to check the availability of license. The issue of licensing health professionals was not the priority of health care managers. The challenges to the poor practice of health professionals' licensing were fraudulent academic credentials, resource shortages, and a weak follow-up and coordination system. Further awareness of licensing practice should be created, especially for lower-level employers. Regulators shall establish a reliable digital system to consistently assure the quality of licensing practice. Health care managers should implement mechanisms to regularly monitor the licensing status of their employees and ensure that government requirements are met. Collaboration and regular communication between regulators and employers can improve quality practice. Further study is needed to explore the licensing practice by participating in other diverse study groups and methodologies.

## Supporting information

**S1 File.**
(DOCX)

**S2 File.**
(DOCX)

## Acknowledgments

The Ministry of Health-Ethiopia and the Ethiopian Public Health Institute deserve our gratitude for allowing us to conduct this national study. We would like to convey our heartfelt gratitude and appreciation to Jhpiego-Ethiopia for their technical support in giving us constructive suggestions and intellectual comments. We would like to express our gratitude to the regulatory staff of the regional health bureaus for their outstanding coordination and administrative actions throughout data collection. Finally, we thank the study participants for taking part in this research process and sharing their experience.

## Author Contributions

**Conceptualization:** Yohannes Molla Asemu.

**Formal analysis:** Eshetu Cherinet Teka, Meron Yakob Gebreyes, Endalkachew Tsedal Alemneh, Biruk Hailu Tesfaye, Yohannes Molla Asemu, Bethlehem Bizuayehu Kebede, Mekonnen Desie Degebasa, Tsedale Tafesse Lemu.

**Investigation:** Eshetu Cherinet Teka, Meron Yakob Gebreyes, Endalkachew Tsedal Alemneh, Biruk Hailu Tesfaye, Ermias Gebreyohannes Wolde, Wondimu Daniel Ashena, Samuel Mengistu, Fikadie Dagnew Biset, Bezawit Worku Degefu, Tangut Dagnew Azeze, Wudasie Teshome Shewatatek, Melese Achamo Seboka, Abera Bezabih Gebreegzi, Yeshiwork Eshetu Abebe, Matias Azanaw Alayu, Fatuma Ahmed Ebrahim, Eden Workneh Sahlemariam, Genet Kifle woldesemayat, Hailemaryam Balcha Admassu, Bethlehem Shikabaw Chekol.

**Methodology:** Eshetu Cherinet Teka, Meron Yakob Gebreyes, Endalkachew Tsedal Alemneh, Biruk Hailu Tesfaye, Firew Ayalew Desta, Yohannes Molla Asemu, Ermias Gebreyohannes Wolde, Wondimu Daniel Ashena, Samuel Mengistu, Tewodros Abebaw Melese, Bezawit Worku Degefu, Bethlehem Bizuayehu Kebede, Tangut Dagnew Azeze, Wudasie Teshome Shewatatek, Melese Achamo Seboka, Abera Bezabih Gebreegzi, Mekonnen Desie Degebasa, Tsedale Tafesse Lemu, Yeshiwork Eshetu Abebe, Matias Azanaw Alayu, Fatuma Ahmed Ebrahim, Eden Workneh Sahlemariam, Genet Kifle woldesemayat, Hailemaryam Balcha Admassu, Bethlehem Shikabaw Chekol.

**Project administration:** Tewodros Abebaw Melese, Fikadie Dagnew Biset.

**Supervision:** Tewodros Abebaw Melese, Bezawit Worku Degefu.

**Visualization:** Yohannes Molla Asemu.

**Writing – original draft:** Eshetu Cherinet Teka, Meron Yakob Gebreyes, Endalkachew Tsedal Alemneh, Biruk Hailu Tesfaye, Firew Ayalew Desta, Yohannes Molla Asemu, Ermias Gebreyohannes Wolde, Wondimu Daniel Ashena, Samuel Mengistu, Tewodros Abebaw Melese, Fikadie Dagnew Biset, Bezawit Worku Degefu, Bethlehem Bizuayehu Kebede, Tangut Dagnew Azeze, Wudasie Teshome Shewatatek, Mekonnen Desie Degebasa, Tsedale Tafesse Lemu, Yeshiwork Eshetu Abebe, Matias Azanaw Alayu, Fatuma Ahmed Ebrahim, Eden Workneh Sahlemariam, Hailemaryam Balcha Admassu, Bethlehem Shikabaw Chekol.

**Writing – review & editing:** Eshetu Cherinet Teka, Endalkachew Tsedal Alemneh, Biruk Hailu Tesfaye, Firew Ayalew Desta, Yohannes Molla Asemu, Ermias Gebreyohannes Wolde, Bezawit Worku Degefu, Bethlehem Bizuayehu Kebede, Melese Achamo Seboka, Abera Bezabih Gebreegzi.

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
