## [Decision Letter · Decision Letter 0]

10 Apr 2023

PONE-D-22-33343Health care managers’ perspectives on workforce licensing practice in Ethiopia: A qualitative studyPLOS ONE

Dear Eshetu Cherinet ,

Thank you for submitting your manuscript to PLOS ONE. After careful consideration, we feel that it has merit but does not fully meet PLOS ONE’s publication criteria as it currently stands. Therefore, we invite you to submit a revised version of the manuscript that addresses the points raised during the review process.

ACADEMIC EDITOR: I agree with the reviewers' comments and advise the authors to address all of them or provide justifications Indicate the contributions of each author based on the ICMJE | Recommendations | Defining the Role of Authors and Contributors here (https://www.icmje.org/recommendations/browse/roles-and-responsibilities/defining-the-role-of-authors-and-contributors.html )Qualitative research studies should be reported in accordance to the Consolidated criteria for reporting qualitative research (COREQ) checklist or Standards for reporting qualitative research (SRQR) checklist. Further reporting guidelines can be found in the Equator Network's Guidelines for reporting qualitative research. Complete and attach a relevant checklist for the qualitative study.Seek language editing support to improve the readability and clarity of the paper.

Please submit your revised manuscript by May 25 2023 11:59PM. If you will need more time than this to complete your revisions, please reply to this message or contact the journal office at plosone@plos.org. Please include the following items when submitting your revised manuscript:

We look forward to receiving your revised manuscript.

Kind regards,

Bereket Yakob, Ph.D.

Academic Editor

PLOS ONE

Journal Requirements:

No specific funding available for this work.

4. Please include your tables as part of your main manuscript and remove the individual files. Please note that supplementary tables (should remain/ be uploaded) as separate "supporting information" files

Reviewers' comments:

Reviewer's Responses to Questions

**Comments to the Author**

1. Is the manuscript technically sound, and do the data support the conclusions?

Reviewer #1: Yes

Reviewer #2: Yes

2. Has the statistical analysis been performed appropriately and rigorously? 

Reviewer #1: No

Reviewer #2: Yes

3. Have the authors made all data underlying the findings in their manuscript fully available?

Reviewer #1: Yes

Reviewer #2: Yes

4. Is the manuscript presented in an intelligible fashion and written in standard English?

Reviewer #1: No

Reviewer #2: Yes

5. Review Comments to the Author

Reviewer #1: This is an interesting paper that would add knowledge to the HRH regulation and healthcare safety.

Please find comments for further development.

Title: the study doesn’t include national-level managers and accreditors. As such, I would add “subnational level”

Authorship: 26 authors are much for such type of articles. Also, check authors’ contributions according to the International Committee of Medical Journal Editors (ICMJE) recommendations.

Abstract: lines 46-47 and 54-55 specify the results and key home-take recommendations

Background

1. The problem statement and rationale of the study is not clear.

2. Paragraphs 1, 3, and 5 are only single sentences that do not fulfil the principle of paragraphing. In addition, please limit your citations to low-and-middle-income countries. As such, you may drop the 3rd paragraph.

3. I would reorganize the Introduction section for smooth flow. Describe the problem statement in the first paragraph; the second paragraph could be about national HRH regulation standards and strategies; the third paragraph would focus on the HRH regulation shortfalls (including HRH shortages); and the fourth paragraph could be knowledge gaps and rationale of this study.

Methods

1) Settings: I would describe the profile of major health cadres of the country for external readers.

2) Any software used for data analysis?

3) Comparative analysis to compare variations as a multiple case study approach amongst HR managers and accreditors would be helpful. For instance, awareness of licensing practices could not be a theme among accreditors

4) Limitations: What would be the limitations of not including non-managers/practitioners’ perspectives in this study? I would reflect on this in the Discussion section

Results

1. Lines 179-189 are about the implementation fidelity of the licensure or implementation challenges. I don’t think it is related to awareness of licensure

2. Lines 206-217, the way presented here is about challenges rather than enforcement mechanisms

3. Lines 237-248 better fits “Systems for assuring the quality of licensing practices”

4. Sub-themes of the “Challenges of licensing practice” are broad and vague, and some of the paragraphs are not directly related. For instance, lines 276-281 are not related to organizational problems. Similarly, lines 301-307 are not also technical challenges. Rather it is a fraud, an individual behavior. I would recategorize the challenges as 1) fraudulent academic credentials, 2) poor coordination and network between the human resource department and regulatory body, 3) shortage of resources (HR, finance, equipment, and supplies), 4) lack of training or SBCC about licensing, and 5) weak support and governance system

Discissions

1. Often in scientific writing, the first paragraph is dedicated to a summary of key findings and then subsequent paragraphs would compare each key finding with previous literature and discuss implications

2. Lines 359-364 and 377-380 looks like repetitions of results and good to drop

References: Check the references again reference # 22, 23, and 31 are not complete

Language: it has multiple grammatical errors that should be fixed; would benefit from experienced editors

Reviewer #2: Workforce licensing practice in Ethiopia: A qualitative study

Date 6 April 2023

Dr. Mengistu Meskele

General comment

This is original research and a very important area of research in the health system quality in Ethiopia. It will befit for policy and program aspects. It is well written and triangulated with mixed methods research, is a strength. However, I have the following few comments.

1. Abstract: The word count should not exceed 300/350, as your abstract word count was 380. I think that exceeds the journal requirement. Also, include keywords at the end of the abstract.

2. In the abstract section, the author has to include the sample size and techniques for qualitative study and the brief scientific rigours (credibility, dependability and transferability ). How they come across to maintain the quality of the study has to be included.

3. In line 48 the heading has to only ‘’conclusion’’ and word recommendation should be removed.

4. Line 51” a reliable system to continuously assure the quality of licensing practice ‘’ Kindly put the reliable system that your research found. Otherwise, it is still broad and does not explicitly indicate the solution to policymakers. What is a reliable system?In line 55, encouraging interventions are recommended to respond to the above-identified challenges’’ has to be removed as it does not indicate the other intervention proposed.

Background

Line: 64 ‘’US’’ I think it is the United States, and all the rest abbreviation has to be written in full at the first encounter and the beginning of the sentences. Consider this comment for all the rest abbreviations. Example Line 80 LMICs

Line 55’’Other

Line 140 -144 has to be moved to the result section as how many themes you have identified.

Line 145: Phenomenological analysis framework??? What is that? Have you used Framework analysis / Collazi phenomenological? Not clear?

Line 115: Under quality control, no scientific rigors are included and maintained in this qualitative research (Trustworthiness, credibility, dependability, reflexibility, transferability). Also, no reference were mentioned from where the author got the data collection tool were developed.

Kindly fill the: Consolidated criteria for reporting qualitative research (COREQ) for this research???

Results

Kindly describe themes after the sociodemographic sections

Discussion: Revisited

Some paragraphs are not supported with body of articles.For example 381-384. Kindly support your discussion points with references from other research.

References

Are not complete, and editorial and typographical errors. URL and Accessed date were not included for the grey literature. For example, Reference numbers: 19,21,20,22,23,27,31 etc, are not correct.

---

## [Author Response · Author response to Decision Letter 0]

27 Jul 2023

All responses are separately attached.

---

## [Decision Letter · Decision Letter 1]

24 Nov 2023

PONE-D-22-33343R1Health care managers’ perspectives on workforce licensing practice in Ethiopia: A qualitative studyPLOS ONE

Dear Dr. Teka,

Thank you for submitting your manuscript to PLOS ONE. After careful consideration, we feel that it has merit but does not fully meet PLOS ONE’s publication criteria as it currently stands. Therefore, we invite you to submit a revised version of the manuscript that addresses the points raised during the review process.

Please see the comments below and submit your revised manuscript by Jan 08 2024 11:59PM.. If you will need more time than this to complete your revisions, please reply to this message or contact the journal office at plosone@plos.org. Please include the following items when submitting your revised manuscript:A rebuttal letter that responds to each point raised by the academic editor and reviewer(s). You should upload this letter as a separate file labeled 'Response to Reviewers'.A marked-up copy of your manuscript that highlights changes made to the original version. You should upload this as a separate file labeled 'Revised Manuscript with Track Changes'.An unmarked version of your revised paper without tracked changes. You should upload this as a separate file labeled 'Manuscript'.

We look forward to receiving your revised manuscript.

Kind regards,

Bereket Yakob, Ph.D.

Academic Editor

PLOS ONE

Additional Editor Comments

General comments

- The manuscript requires thorough grammar and punctuation editing. There are many typos, and it will be difficult for the audience to follow the content. Language editing is highly recommended to ensure clarity and readability.

- List of authors: Confirm the contributions of every author. Follow PLOS One authorship guidelines. You can find it here https://journals.plos.org/plosone/s/authorship#loc-authorship-requirements

Abstract:

- Background: State the problem in the background. It only seems your study was about the importance of professional licensing bodies, not the barriers.

- Please improve the grammar and readability of the text in the abstract. For instance, 1) in the methods, “An interview guide was prepared in English, translated to” this must be changed into “translated into.” 2) results: “awareness on licensing practices” needs to be changed into “awareness of licensing practices.”

- Did you use any software to analyze the qualitative data, or did you do it manually?

- “Lack of awareness among managers was reported especially at lower-level employers.” Lack of awareness of what? Please specify it.

- “While regulators were clear on the requirements, employers placed an unwarranted emphasis on ensuring their employees met government licensing requirements.” This is unclear! Please state it in a simple language.

- “Lack of a quality assurance mechanism was reported.” By whom? Quality assurance of what?

- Reorganize and present the results to the four themes. Now, they are scattered all over.

- Embolden the participants' lived experiences by showcasing not only the challenges but also their positive experiences.

- Conclusions: “This study reported a sub-optimal health professionals’ licensing practice in Ethiopia.” You did not tell us what the optimal licensing practices would be or were that were unpracticed. But here, you are saying it was sub-optimal.

- Some of the recommendations were far-fetched and were not from the study results. For instance, the lack of digital systems and the lack of collaboration and communications between regulators and employers were not in the results.

- What is the distinction between “regulators” and “employers”? Who were the employers? Who were the regulators?

Background:

- General, seek editing services and improve language in the manuscript. Otherwise, it is difficult to follow it.

- Line 74: “The Ministry of Health and Education” – Is/was there any ministry called this way? These are separate entities and must be called by their formal names, such as the Ministry of Health and the Ministry of Education.

- Lines 91-92: “Therefore, this study aimed to explore the practice 92 and challenges of health professionals' licensing among Health care managers in Ethiopia.” The practice was not given due attention in the manuscript. The focus was simply on the challenges. You need to revise the above sentence or ensure sufficient details about the practices are included in the manuscript.

Methods

- Insert references for notable methods borrowed from elsewhere.

- Edit language and address typos.

- Lines 112-114: What was the rationale for selecting 12 regional HR and 12 regional regulatory managers? Why only 4 zonal and 4 district HR managers? What was the rationale for the sampling strategy, and what was the purpose of the purposive sampling? Discuss this further.

- What were the roles of zones and districts regarding licensing and regulation? Discuss this further.

- What was the reason for excluding the national licensing and regulatory bodies from the study? Do you need to revise the study topic accordingly?

- Lines 114-116: “Authors had access to information that could identify individual participants can be identified during or after data collection by their labelling as (regional regulatory manager 1 etc.).” The sentence is unclear! How did you ensure the ethical issues about this? Didn’t this endanger the confidentiality and privacy of the participants? Why did you not anonymize the data before sharing it with all authors? Discuss and correct it if there is a reasonable ground for you to share identifiable data with all authors.

- Line 119: Who prepared the interview guide? How did you ensure its content validity?

- Line 153: “We used thematic content analysis to analyse the result.” Discuss the method in brief and insert references. This must come early when you talk about the methods.

- Did you use any software to assist with handling and analyzing the qualitative data?

- Line 145-146: “Coding was done by 146 eight of the investigators (authors). Codes were given for emerging ideas.” Too many people coded the data! How did you ensure variations in coding and analyzing the data? How did it impact interpretation?

- Line 154: It seems something is misplaced or missing.

- Lines 157-158: “In this study, health care managers were represented by regulatory managers 158 and human resource managers in Ethiopia.” The sentence is unclear!

Results

- Lines 171-172: “The highest proportions 172 (75 %) of participants were from the regional level.” Rephrase the sentence.

- Lines 195-196: Were the themes there, or did they emerge during the analysis and interpretation? How distinct are the themes from each other? For instance, what was the difference between “enforcement of licensing practices” and “Systems for assuring the quality of licensing practices”? Were they not more or less similar?

- Lines 212-244: Nothing here seemed to be “enforcement of the licensing practice.” It was simply procedures the licensing bodies followed to grant licensure to healthcare workers. Define what you meant by “enforcement” and discuss its attributes. Show appropriate exhibits as evidence for it.

- Lines 245-280: What is written here is more about the availability and practices of regulatory mechanisms. However, the data were not synthesized to a higher-level theme. Further analysis, synthesis, and interpretation will be helpful.

- Line 281+: Besides what is said in this section, many challenges are mentioned in lines 212-280. For instance, lack of awareness and orientation for staff on licensure procedures and regulatory mechanisms, unavailability of digital systems, poor planning and execution, etc.” Further synthesis, theme development, and interpretation need to be done.

- Lines 320-324: How has this become a health system problem?

Reviewers' comments:

Reviewer's Responses to Questions

**Comments to the Author**

1. If the authors have adequately addressed your comments raised in a previous round of review and you feel that this manuscript is now acceptable for publication, you may indicate that here to bypass the “Comments to the Author” section, enter your conflict of interest statement in the “Confidential to Editor” section, and submit your "Accept" recommendation.

Reviewer #2: (No Response)

2. Is the manuscript technically sound, and do the data support the conclusions?

Reviewer #2: Yes

3. Has the statistical analysis been performed appropriately and rigorously? 

Reviewer #2: Yes

4. Have the authors made all data underlying the findings in their manuscript fully available?

Reviewer #2: Yes

5. Is the manuscript presented in an intelligible fashion and written in standard English?

Reviewer #2: Yes

6. Review Comments to the Author

Reviewer #2: All comments that has provided to revise during the previous time were well addressed.

7. PLOS authors have the option to publish the peer review history of their article (what does this mean?). If published, this will include your full peer review and any attached files.

Reviewer #2: **Yes: **Mengistu Meskele

---

## [Editor Report · Decision Letter 2]

6 Feb 2024

PONE-D-22-33343R2Health care managers’ perspectives on workforce licensing practice in Ethiopia: A qualitative studyPLOS ONE

Dear Dr. Teka,

Thank you for submitting your manuscript to PLOS ONE. After careful consideration, we feel that it has merit but does not fully meet PLOS ONE’s publication criteria as it currently stands. Therefore, we invite you to submit a revised version of the manuscript that addresses the points raised during the review process.

**ACADEMIC EDITOR: **

Lines 72 – 75: FMHACA was reformed to a different organization. Which organization or health agency was responsible for managing health workers' licensure when the study was done? Discuss how FMHACA (or another organization) and the MOH streamline the licensure practice. Based on your presentation, who was responsible for the task is unclear.

Line 88: Remove the unnecessary parenthesis and full stop.

Lines 100-101: “This study was conducted in Ethiopia, a 101 country in Africa with an estimated population of 110 (15).” Something is missing here – Did you mean 110 million?

Lines 107 – 111: Cite references.

Line 105: Remove “o” at the end of the sentence.

Lines 118-120: Cite references

Lines 122 – 128: How many participants from regions, zones, and districts were included in the study? Mention the names of the regions that participated in the study or justify not mentioning it. If there were good reasons for hiding the details, use codes to represent the regions and other structures and present the data. At least provide summaries of the number of regions, zones, and woredas that participated and the number of people who participated from each region and structure. Align this with Table 1 (Line 204).

Line 128: “We anonym zed the data by removing any identifying…” The word anonymized was broken, and there were typos.

Lines 206 – 359: The results happen to be raw. A high-level conceptualization and synthesis might be useful.

We look forward to receiving your revised manuscript.

Kind regards,

Bereket Yakob, Ph.D.

Academic Editor

PLOS ONE

Journal Requirements:

Additional Editor Comments:

Check typos and address them. There are errors in almost all sections. The manuscript may benefit from proofreading and editing services.

---

## [Author Response · Author response to Decision Letter 2]

22 Mar 2024

The responses to reviewer or editor are attached .

---

## [Editor Report · Decision Letter 3]

28 Mar 2024

Health care managers’ perspectives on workforce licensing practice in Ethiopia: A qualitative study

PONE-D-22-33343R3

Dear Dr. Eshetu,

We’re pleased to inform you that your manuscript has been judged scientifically suitable for publication and will be formally accepted for publication once it meets all outstanding technical requirements.

Kind regards,

Bereket Yakob, Ph.D.

Academic Editor

PLOS ONE

Additional Editor Comments (optional):

Checking and modifying the bibliography may be required.
---

## [Editor Report · Acceptance letter]

8 Apr 2024

PONE-D-22-33343R3 

PLOS ONE

Dear Dr. Alemneh, 

I'm pleased to inform you that your manuscript has been deemed suitable for publication in PLOS ONE. Congratulations! Your manuscript is now being handed over to our production team.

Kind regards, 

on behalf of

Dr. Bereket Yakob 

Academic Editor

PLOS ONE